

# Indomethacin reproducibly induces metamorphosis in *Cassiopea xamachana* scyphistomae

Patricia Cabrales-Arellano[2], Tania Islas-Flores[1], Patricia E. Thomé[1] and Marco A. Villanueva[1]

[1] Unidad Académica de Sistemas Arrecifales, Instituto de Ciencias del Mar y Limnología-UNAM, Puerto Morelos, México
[2] Posgrado en Ciencias del Mar y Limnología-UNAM, Instituto de Ciencias del Mar y Limnología-UNAM, Ciudad de México, México

## ABSTRACT

*Cassiopea xamachana* jellyfish are an attractive model system to study metamorphosis and/or cnidarian–dinoflagellate symbiosis due to the ease of cultivation of their planula larvae and scyphistomae through their asexual cycle, in which the latter can bud new larvae and continue the cycle without differentiation into ephyrae. Then, a subsequent induction of metamorphosis and full differentiation into ephyrae is believed to occur when the symbionts are acquired by the scyphistomae. Although strobilation induction and differentiation into ephyrae can be accomplished in various ways, a controlled, reproducible metamorphosis induction has not been reported. Such controlled metamorphosis induction is necessary for an ensured synchronicity and reproducibility of biological, biochemical, and molecular analyses. For this purpose, we tested if differentiation could be pharmacologically stimulated as in *Aurelia aurita*, by the metamorphic inducers thyroxine, KI, NaI, Lugol's iodine, $H_2O_2$, indomethacin, or retinol. We found reproducibly induced strobilation by 50 μM indomethacin after six days of exposure, and 10–25 μM after 7 days. Strobilation under optimal conditions reached 80–100% with subsequent ephyrae release after exposure. Thyroxine yielded inconsistent results as it caused strobilation occasionally, while all other chemicals had no effect. Thus, indomethacin can be used as a convenient tool for assessment of biological phenomena through a controlled metamorphic process in *C. xamachana* scyphistomae.

## INTRODUCTION

Cnidarian–dinoflagellate symbioses are fundamental components of coral reefs and other tropical ecosystems. The biochemical and molecular mechanisms underlying such symbiotic relationships remain poorly understood although important efforts have been carried out to describe transcription profiles in several cnidarian-dinoflagellate systems (*Weis & Levine, 1996*; *Richier et al., 2008*; *DeSalvo et al., 2010*). Due to the difficulty of establishing appropriate models for the study of coral–dinoflagellate symbiosis, new

Corresponding author
Marco A. Villanueva,
marco@cmarl.unam.mx

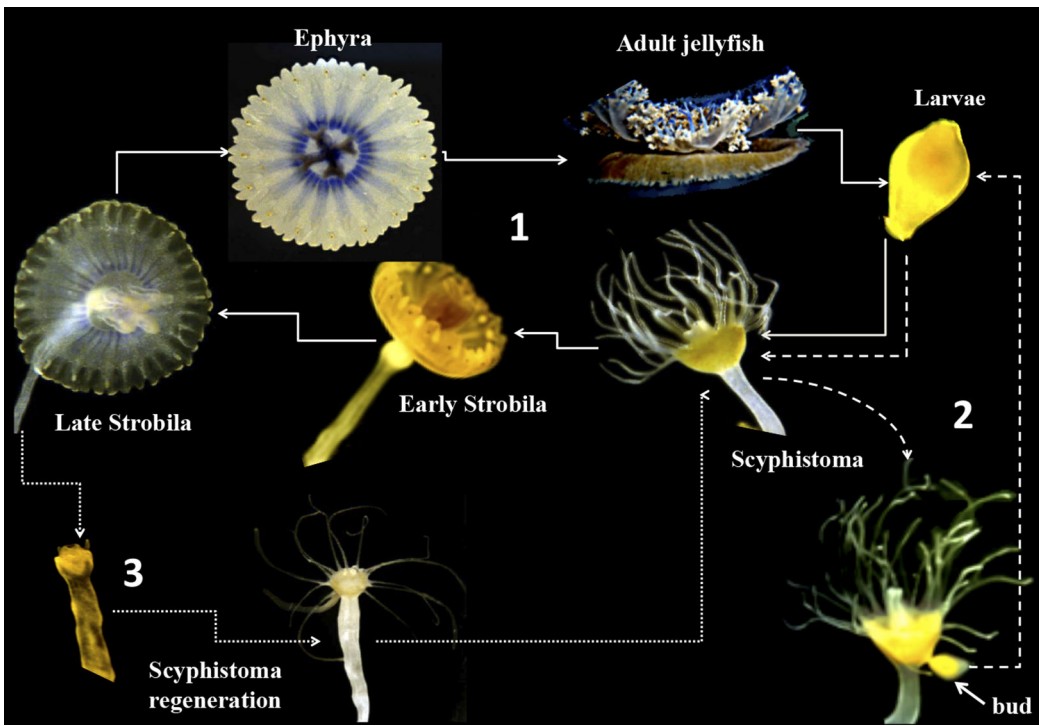

**Figure 1** **Life cycle of *Cassiopea xamachana*.** The cycle starts with sexual reproduction (1, solid lines) when adult jellyfish release their gametes into the water column. There, sperm-fertilized eggs become free-living larval ciliates. Once the swimming larvae identifies a suitable substrate, it settles and develops into a scyphistoma. The final stage is thought to ensue once *Symbiodinium* has been acquired by the scyphistomae, triggering metamorphosis, strobilation and ephyrae formation. The ephyrae are released into the water column creating a free-living jellyfish. In the asexual component (2, dashed lines), the scyphistoma develops a bud that is released into the environment as larva. It settles and metamorphoses to scyphistoma and the cycle perpetuates. In parallel, as the ephyra is released (3), it can regenerate into a newly formed scyphistoma (dotted lines) and enter the asexual part of the cycle.

emerging models such as *Aiptasia pulchella*, *Anemonia viridis* anemonae, and the jellyfish *Cassiopea xamachana* have been used for various biochemical, molecular, and transcriptomics approaches (*Kuo et al., 2004*; *Markell & Wood-Charlson, 2010*; *Moya et al., 2012*). The jellyfish *C. xamachana* offers various advantages for such studies since it can be propagated both sexually and asexually. The sexual cycle occurs when the male and female gametes produce a planula larva, which can settle and metamorphose to a polyp or scyphistoma (*Colley & Trench, 1983*). This scyphistoma can then acquire symbionts and differentiate to an ephyra, which will subsequently become an adult jellyfish (Fig. 1). If the scyphistomae do not acquire the symbiont, they can bud out new larvae, which can settle again and form new scyphistomae to perpetuate the cycle (Fig. 1; *Colley & Trench, 1983*). This physiological process represents an advantage to study the metamorphosis of the jellyfish under controlled laboratory conditions. However, in our hands, we have obtained inconsistent results with the induction of metamorphosis in *C. xamachana* with the infecting symbiont. Furthermore, we have consistently observed symbionts within our asexual scyphistomae cultures, which stay perpetuating the cycle without strobilation or progression to the expected metamorphosis. Since we are interested in studying

signal-transduction processes that occur during the metamorphic process, we required a reproducible and consistent procedure to induce the metamorphosis in *C. xamachana* scyphistomae.

Several compounds have been reported for chemical induction of metamorphosis in jellyfish, mostly *Aurelia aurita*, which does not undergo symbiosis with *Symbiodinium*. These include indomethacin (*Kuniyoshi et al., 2012*), $H_2O_2$ (*Berking et al., 2005*), thyroxine and iodine (*Spangenberg, 1967*, *1974*), retinol, 9-*cis*-retinoic acid and the indole compounds 5-methoxy-2-methyl-3-indoleacetic acid, 5-methoxyindole-2-carboxylic acid, 2-methylindole, and 5-methoxy-2-methylindole (*Fuchs et al., 2014*). One report documenting the use of the iodine-containing compound lugol as an inducer of metamorphosis in *Cassiopea* spp. jellyfish exists (*Pierce, 2005*). In that study, 100% of strobilation was shown to occur after a week of exposure to 0.06 ppm. However, the induction of strobilation in the scyphistomae of this jellyfish with a single defined compound has not been documented.

In this work, we were able to consistently and reproducibly induce metamorphosis in *C. xamachana* scyphistomae by applying a single dose within a range of 0.5–50 $\mu$M indomethacin at 25 $\pm$ 2 °C and 200 $\mu$mol quanta m$^{-2}$ s$^{-1}$ under 12 h light/dark photoperiod cycles. These results place indomethacin as a tool for biochemical and/or molecular studies through a controlled metamorphic process in *C. xamachana* scyphistomae.

# MATERIALS AND METHODS

## Animal rearing

*Cassiopea xamachana* scyphistomae were a kind gift of the Xcaret Park aquarium in Quintana Roo, México. The animals were reared in Petri plates containing filtered seawater and kept at 25 $\pm$ 2 °C under darkness and only exposed to artificial laboratory light when fed. They were fed a diet of live *Artemia salina* nauplii every 2 days and cleaned from debris after feeding.

## Chemicals

Thyroxine, KI, NaI, Lugol's iodine (potassium triiodide), indomethacin, retinol, 9-*cis*-retinoic acid, and dimethyl sulfoxide (DMSO) were from Sigma. $H_2O_2$ was purchased from the local pharmacy.

## Experimental treatments

The animals were stopped from feeding two days prior to exposure to the chemicals. Under fluorescence microscopy, we observed that all the scyphistoma had a few symbionts (Fig. 2). The treatments were carried out under the laboratory artificial ambient light, and when applied, the scyphistomae were placed under a 12-h light/dark cycle with fluorescent lamps at 70 $\mu$mol quanta m$^{-2}$ s$^{-1}$. Five scyphistomae with an average head diameter of approximately 2.5 mm were placed into individual wells of a 6-well microtiter plate with 5 ml sterile artificial seawater (Instant Ocean, Cincinnati, OH, USA) and triplicate wells were used for each experimental treatment. The treatments were as follows: thyroxine

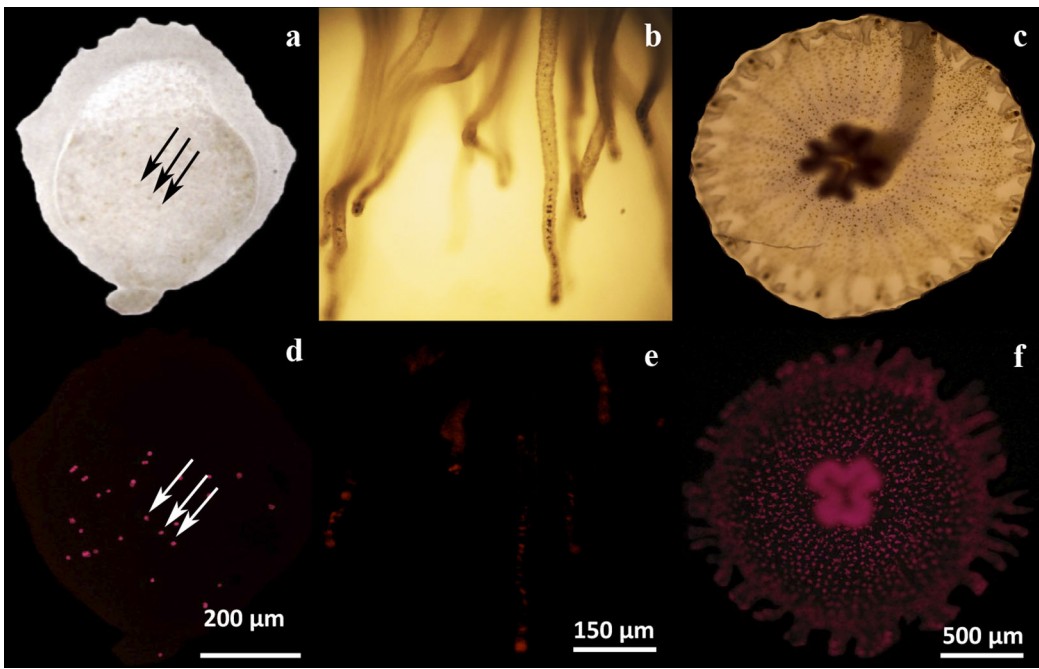

**Figure 2** **Microscopic analysis of *Symbiodinium* presence on three physiological stages of *Cassiopea xamachana*.** Endosymbiotic *Symbiodinium* cells were observed by their contrast against the tissues by light microscopy (A–C), or by their chlorophyll autofluorescence (D–F). Symbionts can be observed as dark or as fluorescent red dots, respectively, in a larval bud (A, D), scyphistoma tentacles (B, E) and strobila (C, F). The arrows clearly show the symbionts as some dark dots (A) corresponding to the same fluorescent ones (D) in a larval bud. Bars show the corresponding dimension references in micrometers.

at 0.1, 1, 5, 10, 20, 50, and 100 $\mu$M; retinol at 0.5, 1, and 5 $\mu$M; 1, 10, and 100 nM $H_2O_2$; 100 $\mu$M glucose; 100 $\mu$M glycine; 50, 100, and 300 $\mu$M L-Tyrosine; 50, 100, and 300 $\mu$M NaI; 100 $\mu$M KI; 0.01% (v/v) glycerol; lugol at 263 $\mu$l/l (equivalent to 130 mg/ml of iodine), and 9-*cis*-retinoic acid at 1 and 25 $\mu$M (*Fuchs et al., 2014*). Indomethacin was tested at 0.5, 1, 5, 10, 25, 50, 100, 200, and 500 $\mu$M. One micromolar of 9-*cis*-retinoic acid according with *Fuchs et al. (2014)* was tested. Controls consisting of filtered seawater or artificial seawater with or without DMSO (as indomethacin was dissolved in DMSO) were also used.

## Microscopy

Induction of metamorphosis to strobilation was monitored visually under a Leica MZ125 (Leica Microsystems) stereomicroscope. In order to monitor for the presence of symbionts inside the various stages of the animals, observations were carried out under a Zeiss Axioskop epifluorescence microscope with a rhodamine filter. Larvae, scyphistomae, or strobilae were previously anesthetized by 10 min incubations with 10% $MgCl_2$ in filtered seawater at $25 \pm 2\,^{\circ}C$, and then placed on the microscope slides for the observations.

## Statistical analysis

Data were statistically analyzed using the R project software (http://www.r-project.org/) with a Nested ANOVA (days within different concentrations of indomethacin) and a Student–Newman–Kleus post hoc analysis.

## RESULTS

### Symbionts are present at various stages of non-strobilating *C. xamachana*

In our hands, asexually reared *C. xamachana* at different physiological stages (maintained in the dark and placed at ambient light only for feeding), consistently showed the presence of symbionts. Larvae were observed to contain endosymbionts detected as dark spots under light microscopy (Fig. 2A, arrows). The same spots showed the characteristic chlorophyll autofluorescence under fluorescence microscopy (Fig. 2D, arrows). Similarly, endosymbionts were also consistently detected in tentacles at the scyphistoma stage under both light (Fig. 2B) and fluorescence (Fig. 2E) microscopy. Even though endosymbionts had been clearly acquired in these two physiological stages, infected scyphistomae did not strobilate and/or differentiate to ephyrae. Comparatively, a strobilating scyphistoma also contained a significant load of endosymbionts (Figs. 2C and 2F). Thus, in our hands, we obtained inconsistent results with the induction of strobilation and metamorphosis in *C. xamachana* with the symbiont. *Thornhill et al. (2006)* reported that when the densities of the *Symbiodinium* reached between 10,000 and 50,000 per scyphistoma, these stimulated the induction of strobilation; but this process could take around 3–5 months. Also, *Rahat & Adar (1980)* evidenced the importance of temperature in the metamorphic process in both symbiotic and aposymbiotic *Cassiopea* scyphistomae; however, this induction was not simultaneous. Therefore, we sought alternative methods to induce a reproducible and synchronous scyphistomae strobilation and subsequent metamorphosis.

### Indomethacin reproducibly induces strobilation

After testing several chemicals in an attempt to induce strobilation in *C. xamachana* scyphistomae (see below), we found a consistent induction with indomethacin whereas no induction was observed when plain seawater or seawater with the vehicle DMSO was used as a negative control (Fig. 3). We tested a range of 0.5–500 μM indomethacin concentrations to induce strobilation. A nested ANOVA analysis indicated significant differences between concentrations (DF = 6, $F = 73.022$, $p = 2.2E^{-16}$) and days within each concentration (DF = 21, $F = 12.889$, $p = 1.57E^{-14}$). A Student–Newman–Kleus post hoc analysis grouped days within each concentration ($p < 0.01$) (Fig. 4). Strobilation of some scyphistomae began on day 5, when the indomethacin concentration was at least 5 μM (Fig. 4, white bar), but it was not uniform and only 50% strobilation was observed at 50 μM concentration at this time (Fig. 4, white bar). After day 6, all scyphistomae began to strobilate within 24 h, and all the indomethacin concentration treatments promoted strobilation (Fig. 4, light gray bar). The indomethacin concentrations of 0.5–5 μM were directly proportional to the percentage strobilation up to the sixth day; however, strobilation became uniform only after the seventh day. Strobilation seemed to induce a spontaneous synchrony of all the strobila since release of ephyrae occurred in all of them at seven days independent of their starting time of strobilation. Thus, the optimum indomethacin concentration for a maximum strobilation induction in a shorter

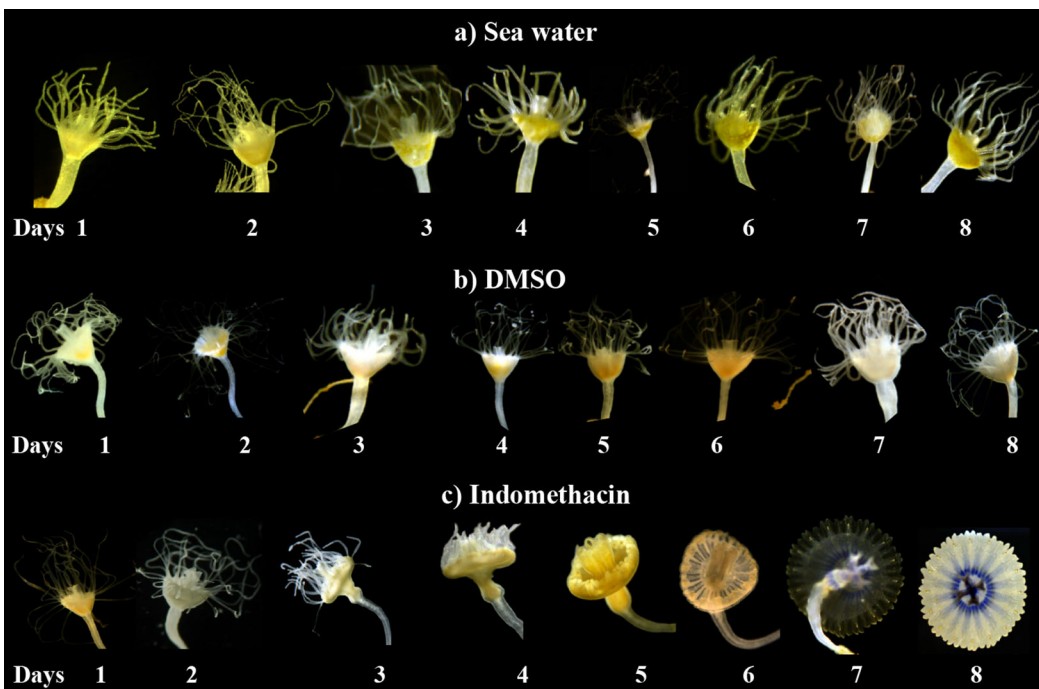

**Figure 3 Induction of strobilation with indomethacin.** Indomethacin (50 μM) was used to induce strobilation on *C. xamachana* scyphistomae. All samples used for the strobilation induction contained symbionts, but only those treated with indomethacin (C) strobilated. Changes can be observed in the calyx of the scyphistomae at day 3, where they begin to show elongation. At day 4, the tentacles start to retract, and at day 5, all the tentacles are absent and the strobila begins pulsating. On day 6 and 7, the ephyra matures, and on day 8, it is released into the environment. In contrast to the indomethacin treatment, the seawater (A) or DMSO (B) vehicle controls did not result in strobilation. The experiment was repeated over three times independently with the same results.

period of time (six days) was 50 μM. Indomethacin at 50 μM also induced strobilation in the dark but the maximum was achieved at 10 days (results exactly the same as 0.5 μM indomethacin in Table S1), indicating that the lack of photoperiod affects the process negatively. In addition, a lower temperature of 22 °C also delayed the strobilation process to 10 days (results exactly the same as 0.5 μM indomethacin in Table S1). These data suggest that this process could be further manipulated by temperature and illumination conditions to accelerate or delay metamorphosis. When indomethacin concentrations higher than 100 μM were tested, they were lethal to the scyphistomae (asterisk in Table S1). It is important to mention that after indomethacin-induced strobilation, the scyphistomae could not be recovered for further asexual propagation.

## Only indomethacin yielded reproducible and consistent results

In addition to indomethacin, we tested glucose, glycine, glycerol, thyroxine, l-tyrosine, KI, NaI, potassium triiodide (Lugol's iodine), $H_2O_2$, retinol, and 9-*cis*-retinoic acid as inducers of metamorphosis in *C. xamachana* scyphistomae under the same temperature and light conditions as indomethacin. We used thyroxine and some iodine chemicals because previous reports documented the use of this hormone and the iodine-based compound lugol to induce strobilation in jellyfish scyphistomae (*Spangenberg, 1974*;

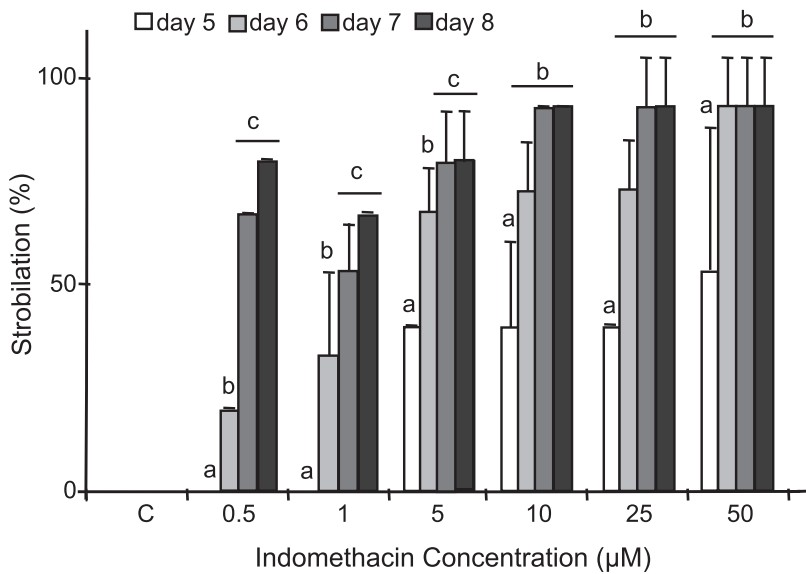

**Figure 4 Induction of strobilation under increasing indomethacin concentrations.** Indomethacin (0.5–50 μM) was used to induce strobilation in *C. xamachana* scyphistomae and percent strobilation recorded after five (white bars), six (light gray bars), seven (dark gray bars), and eight (black bars) days. Triplicate samples each containing five scyphistomae were used for each concentration (see Materials and methods). Experiments were reproducibly performed at least five times. Maximum strobilation within a shortest period of treatment was achieved with 50 μM indomethacin at six days. The bars show the average ± the standard deviation. Post hoc analysis is denoted by small letters at $p < 0.01$.

*Pierce, 2005*). Thyroxine yielded inconsistent results. In all cases, the concentrations were non-lethal but strobilation signs appeared only with 100 μM thyroxine (Table S1) and subsequent ephyrae release occurred only once. On the other hand, 0.5, 1, and 5 μM retinol did not have any effect on the *C. xamachana* scyphistomae and the result was identical as the untreated or mock controls (Fig. 2; Table S1). Conversely, 9-*cis*-retinoic acid was able to induce the strobilation process, but it was slower and not synchronized compared with the indomethacin treatments (Fig. 5). We used two concentrations (1 and 25 μM) for 9-*cis*-retinoic acid, but the highest was lethal (all scyphistomae died). Similarly, glucose, glycine, glycerol, L-Tyrosine, KI, NaI, lugol, and $H_2O_2$ were used at a wide range of concentrations but yielded inconsistent or no induction as well (Table S1).

## DISCUSSION

Indomethacin induction of metamorphosis occurred consistently and in a reproducible manner in *C. xamachana* scyphistomae. The induction was effective at a range of concentrations of 5–50 μM which was within the concentration range observed by *Kuniyoshi et al. (2012)* for *A. aurita* (2.5–20 μM). They reported that, in the case of *A. aurita* induction, the strobilation was dose-dependent, where metamorphosis was induced with the highest doses at nine days and with the lowest ones at 14 days of treatment (*Kuniyoshi et al., 2012*). We obtained similar results in the sense that at 0.5–1 μM strobilation did not occur at five days, whereas it did happen at 5–50 μM. In addition, maximum percent strobilation was achieved at 8 days with 10–50 μM, whereas a statistically significant lower percent strobilation occurred with 1 μM indomethacin

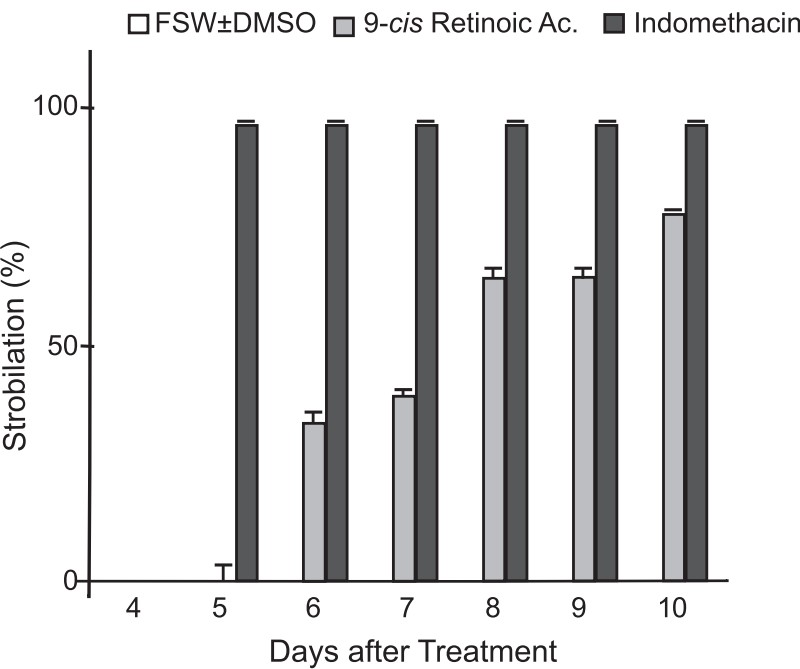

**Figure 5 Comparison of indomethacin and 9-*cis*-retinoic acid effects on strobilation.** Indomethacin at 50 μM and 9-*cis*-retinoic acid at 1 μM were used as inducers for the strobilation of *C. xamachana* scyphistomae, recorded from day 4 to day 10 after each treatment. Triplicate samples containing five scyphistomae each, were followed. Indomethacin consistently induced strobilation from day 5 on (black bars), whereas 9-*cis*-retinoic acid lagged behind even at day 10 (light gray bars). Error bars show the mean average ± standard deviation.

treatment (Fig. 4). Furthermore, strobilation was uniform after the seventh day in the 5–50 μM range. Conversely, thyroxine, which is the protocol inducer for *A. aurita*, yielded inconsistent results as it only caused strobilation occasionally, while all other chemicals had no effect (Table S1). Only 9-*cis*-retinoic acid was also effective at inducing metamorphosis, but at slower times and with an apparent lack of synchronicity. Thus, this compound does not represent a good choice as strobilation inducer for *C. xamachana*.

We do not know through which biochemical mechanism is indomethacin capable of inducing strobilation in *C. xamachana* scyphistomae. Indomethacin is an inhibitor of the cyclooxygenase (COX) enzyme, and therefore of the prostaglandin (PG) biosynthesis; however, when other COX inhibitors (such as aspirin, ibuprofen, etc.) were used, they did not stimulate strobilation in *A. aurita*. Similarly, when the synthesis of arachidonic acid (which is the COX substrate in the PG biosynthesis pathway) was inhibited, strobilation did not occur (*Kuniyoshi et al., 2012*). Thus, the COX pathway of PG biosynthesis does not seem to be the mechanism by which indomethacin induces metamorphosis in these cnidarians. This is also consistent with conflicting results on indomethacin action in mammalian models where it appears to be involved in multiple pathways. For example, indomethacin can inhibit the COX pathway for PG biosynthesis, which is, in turn, synthesized from arachidonic acid (*Smith, Urade & Jakobsson, 2011*). However, in some cases, indomethacin did not inhibit COX expression,

suggesting that there is an alternative COX-independent indomethacin pathway (*Tegeder, Pfeilschifter & Geisslinger, 2001*). Recently, evidence at the proteomic level has suggested the involvement of the Wnt1 signaling pathway without COX activation upon indomethacin treatment in colon cancer cells (*Cheng et al., 2013*). This is consistent with the proposed role of the Wnt1 pathway in cnidarian developmental processes (*Holstein, 2008*). Recently, a peptide hormone with structural similarity to indole strobilation inducer chemicals such as indomethacin has been described as an active molecule to induce strobilation in *A. aurita* (*Fuchs et al., 2014*). Thus, it is likely that indomethacin acts mimicking such peptide hormone action.

## CONCLUSION

This work demonstrates that indomethacin can be used as a reliable chemical inducer of metamorphosis in *C. xamachana* scyphistomae in a consistent and reproducible manner and that this induction may be further manipulated with light and temperature. After the strobilation onset in all scyphistomae, they seem to spontaneously synchronize to produce ephyrae release on the same day. This reproducible chemical induction of strobilation provides a powerful tool for biological, biochemical, and molecular analyses of the metamorphic process under controlled conditions.

## ACKNOWLEDGEMENTS

We thank Claudia Morera, Anthony Rashuam-Cerdán, and Adriana Córdoba-Isunza for technical help. We also thank Luis P. Suescún-Bolívar for help with the statistical analysis and Ana Cerón of Xcaret Park for the scyphistoma donation.

### Funding

The work was funded by grants 175951 from the Mexican National Council of Science and Technology (CONACyT) and IN-210514 from PAPIIT-UNAM. PC-A was supported by PhD fellowship No. 376650 from CONACyT. The funders had no role in study design, data collection and analysis, decision to publish, or preparation of the manuscript.

### Grant Disclosures

The following grant information was disclosed by the authors:
Mexican National Council of Science and Technology (CONACyT): 175951.
PAPIIT-UNAM: IN-210514.
CONACyT: 376650.

### Competing Interests

The authors declare that they have no competing interests.

### Author Contributions

- Patricia Cabrales-Arellano conceived and designed the experiments, performed the experiments, analyzed the data, wrote the paper, and prepared figures and/or tables.

- Tania Islas-Flores conceived and designed the experiments, performed the experiments, analyzed the data, and reviewed drafts of the paper.
- Patricia E. Thomé conceived and designed the experiments, analyzed the data, contributed reagents/materials/analysis tools, prepared figures and/or tables, and reviewed drafts of the paper.
- Marco A. Villanueva conceived and designed the experiments, analyzed the data, contributed reagents/materials/analysis tools, wrote the paper, prepared figures and/or tables, and reviewed drafts of the paper.

## Data Deposition

The following information was supplied regarding data availability.

The raw data has been supplied as Supplementary Dataset Files.

## Supplemental Information

Supplemental information for this article can be found online at http://dx.doi.org/10.7717/peerj.2979#supplemental-information.

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
