# Peer review of "Indomethacin reproducibly induces metamorphosis in Cassiopea xamachana scyphistomae"

_PeerJ, doi:10.7717/peerj.2979_

## Round 0.1 · original submission · Major Revisions

Dear Dr. Villaneuva,

Two reviewers have returned their reviews. Although both deemed the requested revisions to be "minor", I think they are sufficiently major for the reviewers to re-review the manuscript once you have revised it. Please pay particular attention to the comments of reviewer no 2 in regard to the reviewer's questions as to the validity of the findings and the comments for the authors.

Sincerely,
Linda Holland

Reviewer 1 ·

Basic reporting

The authors of the present paper revealed that indomethacin, known as a strobilation-inducing substance in moon jellyfish Aurelia aurita, reproducibly induces metamorphosis in Cassiopea xamachana scyphistomae. The finding provides a useful tool for biological, biochemical, and molecular studies of the metamorphosis in this species.
This work was accomplished well, and the paper is clear and concise for the most part. I would, however, recommend certain potential improvements in the manuscript and Figures.

Some minor comments / suggestions:

> Lines 92 & 93: "Fig. 1"
- Possibly Fig. 2?

> Line 105: "the indole compounds retinol"
- Retinol is not an indole compound.

> Line 176: "Scyphistomae began to strobilate with a maximum difference of only 24 h"
- This description is hard to understand.

> Figure 1:
- Scale bar(s) should be inserted in the photo(s).

> Figure 2:
- Many readers do not know which scyphistoma is, which strobila is, and which ephyra is. The names of stages (planula, scyphistoma, strobila, ephyra, and so on) should be put in Figure 2 or its legend.

> Figure 1 & Figure 2:
- "Life cycle of Cassiopea xamachana" should be shown in Figure 1 and "Microscopic analysis of Symbiodinium presence..." should be Figure 2. The results (microscopic analysis and others) should be shown after the basic information of life cycle is described.

Experimental design

Some minor comments / suggestions:

> Lines 128 - 138: "Experimental treatments"
- In addition to retinol, 9-cis-retinoic acid (9-cis RA) should be tested if possible. Fuchs et al. (2014) showed that both retinol and 9-cis RA have strobilation-inducing activity.

> Line 132: "Five scyphistomae were placed into individual wells of a microtiter plate..."
- The size of microtiter plate should be described (24-well, 12-well, or 6-well plate?). The volume of culture medium should be described.

Validity of the findings

Some minor comments / suggestions:

> Lines 184 & 185: "(not shown)"
- These data should be shown because the relationship between strobilation-inducing activity and temperature / illumination conditions is an interesting topic.

Additional comments

Some minor comments / suggestions:

> Lines 105 & 106: "5-methoxy-2-methyl indole acetic acid", "5 methoxyindole-2 carboxylic acid", "5 methoxy-2-methylindole"
- The names of compounds are incorrect. The correct names are
"5-methoxy-2-methyl-3-indoleacetic acid",
"5-methoxyindole-2-carboxylic acid",
and "5-methoxy-2-methylindole".

> Line 143: "strobile" should be "strobilae".

> Line 309: "strobile" should be "a strobila".

> Line 318: The word "which" should be deleted.

> Line 318: "These" should be "They".

> Line 326: "strobile" should be "strobila".

> Line 328: "Indomethacin" should be "indomethacin".

Reviewer 2 ·

Basic reporting

Conforms to standards

Experimental design

A very interesting manuscript.

Validity of the findings

Did zooxanthellae have anything to do with the strobilation? (the authors said that they often have zooxanthellae-contaminated cultures)
Did the scyphistomae have any symbionts?
What happens if the scyphistomae are induced to strobilate in the dark? (i.e. not a light:dark cycle)

Additional comments

There are some published givens in strobilation of Cassiopeia:
1. they have to be big enough (>1mm diameter polyp?). How big were the polyps in the current study?
2. >22-23oC
3. Some light (not complete darkness)
4. About 5-10,000 zooxanthellae (but see Rahat & Hofmann Endocytobidology III, Vol. 503, New York Academy of Sciences)

---

## Round 0.2 · accepted · Accept

Dear Dr. Villanueva,

I apologize for addressing you by your first name. This is done automatically by the journal as is the letter of acceptance. I cannot change it, only add to it. The written English of your manuscript is adequate. Even so, I suggest that the next time you submit an article, you have it read by a native English speaker.

Sincerely,
Linda Holland

Reviewer 1 ·

Basic reporting

I have confirmed that the authors responded adequately to all my comments. I would like to recommend that the article be acceptable for the PeerJ.

Experimental design

No comment.

Validity of the findings

No comment.

Reviewer 2 ·

Basic reporting

Everything seems fine.

Experimental design

Everything seems fine.

Validity of the findings

Everything seems fine.

Additional comments

Everything seems fine.

---

## Author Rebuttal · Round 0.2

# INSTITUTO DE CIENCIAS DEL MAR Y LIMNOLOGIA
## UNIDAD ACADEMICA DE SISTEMAS ARRECIFALES

UNIVERSIDAD NACIONAL AUTONOMA DE MEXICO

DIRECCIÓN
PROLONG. AV. NIÑOS HÉROES S/N
77580 PUERTO MORELOS, QUINTANA ROO, MÉXICO

APARTADO POSTAL 1152
77580 CANCÚN, Q. ROO; MÉXICO

December 20th, 2016

Dear Editor:
Please find enclosed the revised manuscript entitled "Indomethacin reproducibly induces metamorphosis in *Cassiopea xamachana* scyphistomae" by Cabrales-Arellano P. et al., for publication in PeerJ.

This new version addresses all issues raised by the reviewers, which we also answer on a point-by-point basis below. We hope this revised version is now appropriate for publication.

Reviewer #1: Reviewer 1
Basic reporting
The authors of the present paper revealed that indomethacin, known as a strobilation-inducing substance in moon jellyfish Aurelia aurita, reproducibly induces metamorphosis in *Cassiopea xamachana* scyphistomae. The finding provides a useful tool for biological, biochemical, and molecular studies of the metamorphosis in this species.

This work was accomplished well, and the paper is clear and concise for the most part. I would, however, recommend certain potential improvements in the manuscript and Figures.

Some minor comments / suggestions:

> Lines 92 & 93: "Fig. 1"
- Possibly Fig. 2?
A: This was indeed mixed in the text but now figures were rearranged based on the comment on "Figure 1 & Figure 2" below, so it was corrected to read "Fig. 1" (lines 96 and 98).
> Line 105: "the indole compounds retinol"
- Retinol is not an indole compound.
A: This has been fixed (line 110).
> Line 176: "Scyphistomae began to strobilate with a maximum difference of only 24 h"
- This description is hard to understand.
A: It has been changed to "After day 6, all scyphistomae began to strobilate within 24 h" (line 192).
> Figure 1:
- Scale bar(s) should be inserted in the photo(s).
A: Now placed as Figure 2; scale bars have been added

> Figure 2:
- Many readers do not know which scyphistoma is, which strobila is, and which ephyra is. The names of stages (planula, scyphistoma, strobila, ephyra, and so on) should be put in Figure 2 or its legend.
A: Now placed as Figure 1; labels have been included.
> Figure 1 & Figure 2:
- "Life cycle of *Cassiopea xamachana*" should be shown in Figure 1 and "Microscopic analysis of *Symbiodinium* presence..." should be Figure 2. The results (microscopic analysis and others) should be shown after the basic information of life cycle is described.
A: We agree; they have been changed accordingly.

Experimental design
Some minor comments / suggestions:

> Lines 128 - 138: "Experimental treatments"
- In addition to retinol, 9-cis-retinoic acid (9-cis RA) should be tested if possible. Fuchs et al. (2014) showed that both retinol and 9-cis RA have strobilation-inducing activity.
A: We included an experiment with 9-*cis*-RA and the results compared with the indomethacin treatment. This is shown on a new figure (Figure 5) and results described on lines 219-222.
> Line 132: "Five scyphistomae were placed into individual wells of a microtiter plate..."
- The size of microtiter plate should be described (24-well, 12-well, or 6-well plate?). The volume of culture medium should be described.
A: The information has been added (lines 139 and 141).

Validity of the findings
Some minor comments / suggestions:

> Lines 184 & 185: "(not shown)"
- These data should be shown because the relationship between strobilation-inducing activity and temperature / illumination conditions is an interesting topic.
A: All "data not shown" has been condensed on Supplementary Table 1.

Comments for the author
Some minor comments / suggestions:

> Lines 105 & 106: "5-methoxy-2-methyl indole acetic acid", "5 methoxyindole-2 carboxylic acid", "5 methoxy-2-methylindole"
- The names of compounds are incorrect. The correct names are
"5-methoxy-2-methyl-3-indoleacetic acid",
"5-methoxyindole-2-carboxylic acid",
and "5-methoxy-2-methylindole".
A: They have been corrected (lines 111 and 112).

> Line 143: "strobile" should be "strobilae".
A: It has been corrected (line 154).
> Line 309: "strobile" should be "a strobila".
A: It has been corrected (line 353).
> Line 318: The word "which" should be deleted.
A: It has been deleted (line 345).
> Line 318: "These" should be "They".
A: The plural was changed to singular (line 345).
> Line 326: "strobile" should be "strobila".
A: It has been changed (line 362).
> Line 328: "Indomethacin" should be "indomethacin".
A: It has been changed (line 363).

Reviewer 2
Basic reporting
Conforms to standards
Experimental design
A very interesting manuscript.
Validity of the findings
>Did zooxanthellae have anything to do with the strobilation? (the authors said that they often have zooxanthellae-contaminated cultures)
A: They did not and we clearly stated so when mentioning "Even though endosymbionts had been clearly acquired in these two physiological stages, infected scyphistomae did not strobilate and/or differentiate to ephyrae." (lines 169-171).
>Did the scyphistomae have any symbionts?
A: They do as it is clearly shown on Figures 2b and 2e and mentioned in the text on lines 165-167.
>What happens if the scyphistomae are induced to strobilate in the dark? (i.e. not a light:dark cycle)
A: They did strobilate but he process was delyed as clearly stated on lines 199-202 and shown on Supplementary Table 1.

Comments for the author
There are some published givens in strobilation of Cassiopeia:
1. they have to be big enough (>1mm diameter polyp?). How big were the polyps in the current study?
A: They were 2.5 mm average as stated now on lines 139-140.
2. >22-23oC
A: Temperature was 25ºC (line 156).
3. Some light (not complete darkness)
A: See answer to last comment of Reviewer 1.
4. About 5-10,000 zooxanthellae (but see Rahat & Hofmann Endocytobidology III, Vol. 503, New York Academy of Sciences)

A: Zooxanthellae did not appear to influence the strobilation process; same answer to the first comment.

In addition, we added the two new references Rahat and Adar 1980, and Thornhill et al. 2006.

I am looking forward to receiving the final recommendations.

**Sincerely,**

**DR. MARCO A. VILLANUEVA**